# Dietary Strategies by Foods with Antioxidant Effect on Nutritional Management of Dyslipidemias: A Systematic Review

**DOI:** 10.3390/antiox10020225

**Published:** 2021-02-03

**Authors:** Isabel Medina-Vera, Lizzette Gómez-de-Regil, Ana Ligia Gutiérrez-Solis, Roberto Lugo, Martha Guevara-Cruz, José Pedraza-Chaverri, Azalia Avila-Nava

**Affiliations:** 1Departamento de Metodología de la Investigación, Instituto Nacional de Pediatría, Ciudad de Mexico 04530, Mexico; isabelj.medinav@gmail.com; 2Hospital Regional de Alta Especialidad de la Península de Yucatán, Mérida 97130, Yucatán, Mexico; gomezderegil@gmail.com (L.G.-d.-R.); ganaligia@gmail.com (A.L.G.-S.); roberto.lugo.gomez@gmail.com (R.L.); 3Departamento de Fisiología de la Nutrición, Instituto Nacional de Nutrición y Ciencias Médicas Salvador Zubirán, Ciudad de Mexico 14080, Mexico; marthaguevara8@yahoo.com.mx; 4Departamento de Biología, Facultad de Química, Universidad Nacional Autónoma de Mexico, Ciudad de Mexico 04510, Mexico; pedraza@unam.mx

**Keywords:** dyslipidemias, foods, antioxidants, oxidative stress, inflammation

## Abstract

Nutrition plays a fundamental role in the prevention and treatment of dyslipidemias and its oxidative-related complications. Currently, there is evidence about the beneficial effects of isolated antioxidants or foods enriched or added with antioxidant compounds. However, the application of the natural foods is more integrated than the analysis of a single nutrient. Our aim is compiling scientific literature regarding the nutritional strategies by foods with antioxidant effect in blood lipids, enzymatic and non-enzymatic antioxidants, and oxidative and inflammatory markers of subjects with dyslipidemia. We searched in MEDLINE/PubMed, Scopus, and Web of Science. From a total of 263 studies screened, 16 were included. Dietary strategies included walnuts, olive oil, raw almonds, *G. paraguayase*, white sesame, mate tea, Brazil nut flour, red wine, granulated Brazil nuts, grapes, wolfberry fruit, fermented beverage, coffee, orange, and blackberry juices showed significant differences in blood lipids, antioxidant activity, antioxidant enzymes, and oxidative and inflammatory markers. This systematic review compiling scientific studies about dietary strategies using foods with antioxidant effect to improve the antioxidant status in dyslipidemias.

## 1. Introduction

According to the World Health Organization (WHO), nutrition is one of the highest priorities in public health programs [1]. Scientific evidence has consistently reported the benefits of nutritional strategies to improve global public health [2]. This has favored the acknowledgment of the fundamental role that nutrition plays in the prevention, prognosis, and treatment of several high incidence pathologies, such as dyslipidemia, also related to obesity and type 2 diabetes.

Dyslipidemia is a metabolic alteration characterized by elevated fasting blood levels of total cholesterol (TC), low-density lipoprotein cholesterol (LDL-C), triglycerides (TG), and reduced levels of high-density lipoprotein cholesterol (HDL-C). Dyslipidemia is known as a risk factor to develop insulin resistance, endothelial dysfunction, hypertension and mainly, cardiovascular disease (CVD) [3]. The molecular mechanism that linking dyslipidemia to CVD is directly associated with oxidative stress (OS), this mechanism occurs through multiple processes related with the production of reactive oxygen species (ROS) and inflammation [4]. An excessive amount of ROS increases oxidative damage that in turn, triggers the generation of end products of lipoperoxidation such as malondialdehyde (MDA) and oxidized-low-density lipoprotein (oxidized-LDL). High levels of MDA and oxidized-LDL have been associated with generation of atherogenic plaque and cardiovascular complications [5], which are enhanced due to a decrease of an endogenous antioxidant system either enzymatic including catalase (CAT), superoxide dismutase (SOD), and glutathione peroxidase (GPx), and/or non-enzymatic antioxidants including Oxygen Radical Absorbance Capacity and Ferric Reducing Antioxidant Power values, vitamins C and E, and reduced glutathione content [6]. Furthermore, the OS is also associated with an increased level of pro-inflammatory cytokines such as interleukin-1 (IL-1), interleukin-2 (IL-2), interleukin-6 (IL-6) thromboxane (TXB), C-reactive protein (CRP), and upregulation of inflammatory molecules like vascular cell adhesion molecule-1 (VCAM-1), and intercellular adhesion molecule-1 (ICAM-1). Altogether, excessive ROS, lipoperoxidation products, decrease in antioxidants enzymes and pro-inflammatory cytokines, establish a pro-atherogenic environment and generate endothelial dysfunction [7]. These risks factors associated with the etiology and development of CVD found in dyslipidemias can be modifiable by changes in eating habits [8]. Currently, the global approach of strategies to decrease dyslipidemias includes a nutrition treatment combined with exercise, and also the supplementation of bioactive compounds; this added to medical visits with multidisciplinary teams that include doctors, nurses, and dieticians that results in the improvement of lipid profiles and cardiovascular protection [9].

Many studies provide evidence of an antioxidant effect in dyslipidemias by the consumption of berries [10], garlic [11], green tea [12], soy [13], and cocoa [14]. However, it must be mentioned that the referenced nutritional strategies use food enriching with antioxidants compounds either as isolated antioxidants compounds (e.g., polyphenols, flavonoids, vitamins, and beta-carotenes) [15]. These antioxidants compounds can be found in a variety of natural foods that also provide more nutrients and are also more accessible.

Based on the high prevalence of dyslipidemias and the call for nutritional interventions to attend public health priorities, this systematic review aimed at compiling scientific literature regarding the nutritional strategies by foods with antioxidant effect in blood lipids, enzymatic and non-enzymatic antioxidants and oxidative and inflammatory markers of subjects with dyslipidemia. We hypothesized that nutritional strategies of foods with antioxidant effect will have a positive impact on blood lipids, enzymatic, and non-enzymatic antioxidants, as well as a decrease in oxidative and inflammatory markers in subjects with dyslipidemias. Thus, this review summarized evidence on the issue by including only natural foods, that is, foods that have not been modified by addition of isolated antioxidants compounds. Results can be of interest to those health professionals working with patients with dyslipidemia and to those affected by this pathology. The information might be useful for designing nutritional strategies with foods that are easily accessible and at a lower cost.

## 2. Materials and Methods

The present study was performed following the Preferred Reporting Items for Systematic Reviews and Meta-Analysis (PRISMA) guidelines [16]. Due to the study design, neither Institutional Review Board (IRB) approval nor patient informed consent was required.

### 2.1. Search Strategy

Two authors (AAN and IMV) performed the search strategy independently. The studies were identified through the online sources of MEDLINE/PubMed, Scopus, and Web of Science. The grey literature was searched using the website Google Scholar. The term ‘antioxidant’ was introduced along with the terms ‘food’, ‘dyslipidemia’, ‘human’ and ‘clinical trials’, and ‘intervention’. Here, the antioxidant effect refers to antioxidant activity in plasma or serum, endogenous antioxidant system, and the modulation of oxidative and inflammatory markers. The search was conducted for articles published until December 2019. The description of the Population, Intervention, Control, Outcomes (PICO) strategy [17] applied in the present systematic review; it is based on Subjects with dyslipidemias (Population), Dietary interventions including foods with antioxidant effect (Intervention), Any comparator or comparison, Placebo, Control diet or Without comparator (Control) and Blood lipids, Antioxidant activity, Oxidative and inflammatory markers (Outcomes) (Table 1).

### 2.2. Selection of Studies

After removing duplicates, the same authors screened independently the titles and abstracts for eligibility evaluation, based on the inclusion criteria. Articles that could not be eliminated by title and abstract were retrieved in full and subsequently assessed against the eligibility criteria prior to final inclusion. The same two researchers who evaluated the eligibility criteria they carried out data extraction and quality assessment of the full texts.

### 2.3. Selection Criteria

Original studies were included if they met the following criteria: (1) performed on subjects with dyslipidemia, and/or hypercholesterolemia, and/or hypertriglyceridemia, (2) reporting dietary interventions that included natural foods with antioxidant effect, (3) a design with any comparator or comparison (including comparing to baseline), and (4) reporting outcome in antioxidant activity and/or oxidative and inflammatory markers and/or blood lipids. Exclusion criteria were: (1) participants with dyslipidemia and other pathologies including type 2 diabetes, kidney disease, and/or human immunodeficiency virus (HIV), (2) total or partial children or adolescent participants, and (3) the intervention included food enriched with antioxidants or isolated antioxidants compounds. Studies were also excluded if they contained overlapping subjects with other studies.

### 2.4. Data Extraction

Data extraction from all selected articles were extracted independently by AAN and IMV, which included (1) first author’s name; (2) year of publication; (3) study design; (4) study population; (5) study location; (6) follow-up; (7) number of participants; (8) dietary intervention; and (9) main findings. In case of discrepancy between AAN and IMV, this was resolving by the opinion of another person (LG, ALGS, RL, MGC, and/or JPC).

Excluding duplicates, 263 records were identified, and later 215 were removed after screening titles and abstracts. Most of the studies eliminated at this stage referred to records including participants with type 2 diabetes, kidney disease, or HIV, or with pediatric samples. A total of 48 records were retained for full text examination, 32 were later removed due to the inclusion of supplements, ethanolic extracts, enriched food, or isolated bioactive compounds. Finally, we included 16 records that were read thoughtfully to obtain the following information (Figure 1).

### 2.5. Quality Assessment

Additionally, the risk of bias was assessed with the Cochrane tool [18]. The following items were used: adequacy of sequence generation and allocation concealment to evaluate selection bias, blinding of participants and personnel to evaluated performance bias, blinding of outcomes assessment to evaluated detection bias, incomplete outcome data to evaluate attrition bias, selective outcome reporting to evaluate reporting bias and other probable sources of bias; the quality of the studies was assessed with the CONSORT statement [19]. These assessments were performed independently by AAN and IMV; in the occurrence of discrepancies these were resolved through discussion with a third author (LG, ALGS, RL, MGC, and/or JPC).

## 3. Results

### 3.1. Characteristics of Study Design and Population

The studies included in the analysis showed a ranged of publication from 2004 to 2019 (Table 2). The studies included both men and women, except for Jalali-Khanabadi et al. [20] which only included men. The metabolic conditions of subjects included in the studies were moderate hypercholesterolemia [21,22], mildly dyslipidemia [23], hyperlipidemia [20], hypercholesterolemia [24,25,26,27], hiperlipidemia [28], dyslipidemia [29], mildly hypercholesterolemia [30,31], dyslipidemia with hypertension [32,33,34], and healthy subjects as a control group [25,27]. The studies designs included seven randomized trial [26,28,29,30,31,34], six with randomized crossover trial [21,23,25,27,33], and three evaluation of baseline and post intervention [20,24]. The studies were conducted in Brazil [29,32,33,35], Iran [20,26,28,34], Spain [21,27], China [24,31], Korea [30], Greece [25], United Stated of America [22], and Italy [23]. Six studies were interventions coupled with washout period or diet standardization [21,22,23,27,32,33]. 

### 3.2. Dietary Interventions

Different fruits, vegetables, and nuts are a source of antioxidants that can reduce the risk of oxidative-related diseases. The included studies based their interventions on foods such as raw almonds [20], walnuts [21,22], olive oil [23], *Graptopetalum paraguayense* [24], red wine [25], grapes [26], coffee [27], white sesame seeds [28], mate tea [29], wolfberry fruit [30], and fermented beverage made from fruits and vegetables [31], Brazil nut flour [32], granulated Brazil nut [33], orange juice and blackberry juice [34], which have been described as the most antioxidant-rich foods [36]. Due to the chemical composition of the foods included in the studies, the main effects observed were in antioxidant activity, oxidative and inflammatory markers, and on the blood lipids (Table 2).

### 3.3. Effects of Interventions with Foods on Antioxidant Activity

Dyslipidemias are highly related with excessive reactive oxygen species (ROS) generation, which in turn are associated with development of cardiovascular disease (CVD). An outcome of interest is the antioxidant effect and its potential impact on the management of dyslipidemias. Of the sixteen selected studies, eleven reported an antioxidant effect from the intervention [22,24,25,26,27,28,29,30,31,33,35]. Seven studies showed an increase on antioxidant activity in plasma or serum produced by the dietary interventions [22,25,26,27,29,31,35].

One of these studies showed that consumption of 750 mL/day of orange juice significantly increased antioxidant activity after 8 weeks of intervention. Interestingly, studies that included a placebo or control group showed a significant antioxidant effect only with consumption of foods with antioxidants such as oil and skin nuts [22], red wine [25], red and white grapes [26], coffee [27], Mate tea [29], fermented beverage made from plant materials including vegetables, fruit, seaweed and processed food, herbal products from different plant materials; also it contains yeast, plum extract, brown sugar, raw sugar, maltose, glucose, fructose, sucrose, and oligosaccharides like inulin and fructo-oligosaccharides [31]. In addition, four studies also observed an effect in the expression and/or activity of endogenous antioxidant enzymes [24,28,30,33]. One of these studies, using the *G. paraguayense*, a popular traditional Chinese herb that is consumed as a vegetable, showed an increase in the enzymatic activity of GPx (39.3 ± 6.6 to 45.8 ± 4.9 units/mg protein, *p* = 0.02) and CAT (4.6 ± 0.5 to 5.3 ± 0.4 units/mg protein, *p* < 0.001) [24]. As well, aqueous wolfberry fruit consumption significantly enhanced the activity of CAT [30]. Furthermore, sesame seeds intake promote an increased in the activities of SOD (1754.9 ± 269.9 to 1734.4 ± 426.6, *p* < 0.05) and GPx (21.4 ± 2.2 to 21.7 ± 2.4, *p* < 0.05), and without differences in the control group [28]. A granulated Brazil nut intervention showed an increased in GPx3 activity (112.66 ± 40.09 to 128.32 ± 38.31 nmol/min/mL, *p* < 0.05) [33]. In addition, two studies showed the effect on concentration of non-enzymatic antioxidants [24,29]. The intervention with *G. paraguayense* promoted an increase of glutathione (GSH) from 138.7 ± 41.7 to 179.7 ± 20.7 (*p* = 0.02) [24]. A similar effect was shown by consumption of Mate tea, which significantly increased the GSH blood concentration from 83.8 ± 2.1 to 102.0 ± 3.1 μmol/L (*p* < 0.05) [29].

Antioxidant activity is also a function of the individual and synergistic effects of numerous bioactive compounds, and its interaction with endogenous enzymatic antioxidants. Granulated Brazil nut intake increased plasma selenium from 87.0 ± 16.8 to 180.6 ± 67.1 μg/L (*p* < 0.05) [33], this is an important component in the antioxidant function of selenoproteins such as GPx. Another compound that enhance antioxidant activity are ascorbic acid, tocopherols, and vitamins. After 8 consecutive weeks of *G. paraguayense* consumption, there was a significant increase in plasma concentration of ascorbic acid from 6.4 ± 1.1 to 9.2 ± 2.8 μmol/L (*p* = 0.01), and α-tocopherol from 3.6 ± 0.6 to 4.1 ± 0.6 μmol/L (*p* = 0.04) [24]. Additionally, intervention with red wine showed similar results, after its consumption increased the concentration of α-tocopherol by 13.1% (*p* = 0.002).

### 3.4. Antioxidant Effect of Foods on Oxidative and Inflammatory Markers

Oxidative markers are mainly thiobarbituric acid reactive substances (TBARS), MDA, and oxidized-LDL result from lipoperoxidation induced by ROS. Notably, the effects observed in vivo by dietary interventions that include antioxidant foods include the reduction of oxidative damage in lipids, as was demonstrated by seven of the reviewed studies. Two studies were reported decrement in TBARS levels compared to baselines, one providing participants with a daily intake of 40 g of sesame seeds for 60 days (34.5%) [28], and another study making participants consume for 8 weeks 60 mL/day of a fermented beverage made from fruits or vegetables (23%) [31]. Additionally, TBARS decreased by 23% (*p* < 0.001) after consumption of red grape and by 6% (*p* = 0.02) after consumption of white grape after an eight-week intervention period compared to baseline [26]. Furthermore, MDA decreased by consumption of orange juice by 55% [35], *G. paraguayense* by 28.1% [24] and coffee by 10.3% [27]. The granulated Brazil nut intake significant decreased oxidized-LDL (66.31 ± 23.59 to 60.68 ± 20.88 U/L, *p* < 0.05) [33].

Inflammation is also triggered in dyslipidemias associated with OS. Interestingly, food with antioxidant effect has demonstrated to reduce anti-inflammatory markers such as TXB, CRP and VCAM-1. Serum TXB2, a marker of production by maximally activated platelets, decreased 20% after olive oil consumption [23]. In addition, other interventions that promoted anti-inflammatory effects by decreasing CRP concentration after consumption of orange and blackberry juices [34]. Furthermore, an anti-inflammatory effect was observed by walnuts intake with a 20% decrement in the VCAM-1 concentration, a molecule related to the presence of atherosclerotic lesions [21].

### 3.5. Effect of Foods with Antioxidant Effect on Blood Lipid Levels

Nine studies reported a significant effect on blood lipid concentration after the dietary interventions with food with antioxidants [20,21,26,27,28,31,32,34]. In one of these studies, the participants were randomized in a crossover design using two diets: Mediterranean-type diet as control and an isoenergetic diet enriched with walnuts. Dietary intervention by replacing 32% of monounsaturated fatty acids energy with walnut significantly reduced TC by −4.4 ± 7.4% (from 6.93 ± 0.70 to 6.43 ± 0.69 mmol/L) and LDL-C in −6.4 ± 10.0% (4.75 ± 0.62 to 4.33 ± 0.47 mmol/L) (*p* < 0.05) when compared with a control group that followed an intervention with Mediterranean-type diet [21]. Similarly, the sesame seed intervention with 40 g white sesame seeds daily for 60 days decreased TC (241.2 ± 41.2 to 221.5 ± 45.2 mg/dL), LDL-C (159.7 ± 37.8 to 144.0 ± 43.7 mg/dL) and TC/HDL-C ratio (3.5 ± 1.0 to 3.1 ± 1.1) (*p* < 0.05) when is compared with a diet without sesame seeds [28]. The intervention with different types of grapes, two groups were randomized to receive 500 g of Condori red grapes or 500 g of Shahroodi white grapes (in 5 serving of 100 g for 8 weeks; grape red intervention diminished TC from baseline 242.61 ± 4.97 to 220.85 ± 7.2 mg/dL (*p* < 0.01), white grape decreased baseline from 230.45 ± 30.54 to 211.4 ± 33.73 mg/dL (*p* < 0.01), and the control group that consumed 5 servings of other fruits except grapes had no shown changes (from 231.57 ± 27.03 to 228.71 ± 28.72). Respect to LDL-C concentration, intervention with red and white grapes decreased its concentration from 164.9 ± 20.92 to 140.61 ± 32.36 mg/dL and 147.2 ± 37.68 to 132.95 ± 35.43 respectively (*p* < 0.05); in contrast, the control group did not show changes in LDL-C concentration (from 149 ± 29.10 to 153.2 ± 28.04 mg/dL). However, only with red grapes intervention was there a significant reduction of TC concentration (−24.22 ± 29.95) compared to the control group (−2.85 ± 28) (*p* < 0.05) [26].

In contrast, blackberry juice increased levels of HDL-C from baseline (38.25 ± 8.75 to 43.11 ± 7.41 mg/dL) compared to the control group who consumed a usual diet (41.55 ± 8.81 to 43.29 ± 7.24 mg/dL). Interestingly, there was not changes for TC after blackberry intervention (from 224.47 ± 25.05 to 223.19 ± 32.57) compared with control group (from 224. 75 ± 30.85 to 226.06 ± 38.83) (*p* = 0.84); but for apolipoprotein A-1 (Apo A-I) concentration they observed an increase from baseline (142.14 ± 24.52 to 149.44 ± 21.677 mg/dL, *p* = 0.015) compared to the control group (149.25 ± 23.64 to 144.17 to 24.12 mg/dL, *p* = 0.11). Additionally, there was a significant difference between changes in ApoA-1 concentration (7.30 ± 17.09 vs. −5.08 ± 19.10, *p* = 0.0005) [34]. On the other hand, partially defatted Brazil nut flour intake (13 g daily for 90 days) significantly decreased TC, non-HDL-C, and Apo A-1 (−20.5 ± 61.2, −19.5 ± 61.2, and −10.2 ± 26.7; respectively) (*p* < 0.05) compared with placebo group that consumed dyed cassava flour (−7.4 ± 44.5, −8.2 ± 44.5, and −7.9 ± 27.8; respectively) [32]. Consumption for 8 weeks of fermented beverage made from plants significantly decreased TC in the fermented plant group (230.05 ± 27.73 to 211.55 ± 31.97 mg/dL) compared with placebo group (226.68 ± 29.22 to 220.18 ± 24.91 mg/dL) (*p* < 0.01). Additionally, decreased LDL-C concentration in the fermented plant group (146.60 ± 14.18 to 130.69 ± 17.71) compared with control group (142.52 ± 13.60 to 138.67 ± 12.92 mg/dL) (*p* < 0.05). Placebo in control group was a beverage that contained plum extract, brown sugar, raw sugar, maltose, glucose, fructose, and sucrose [31]. A similar effect was observed with a consumption of 6 g/day of soluble green (35%)/roasted (65%) coffee intervention that showed a significant reduction in TC (231.4 ± 4.9 to 210.4 ± 4.8 mg/dL, *p* = 0.006), LDL-C (154.7 ± 4.1 to 135.9 ± 5.1 mg/dL, *p* = 0.001) and TG (103.3 ± 7.5 to 82.9 ± 6.1 mg/dL, *p* = 0.017) [27]. The interventions with 60g of raw almond consumption [20] significantly decreased LDL-C (169.11 ± 27.03 to 145.17 ± 25.48 mg/dL) (*p* < 0.001), TC (255.21 ± 26.25 to 231.27 ± 39.77 mg/dL) (*p* = 0.01) and apo-B100 (1.26 ± 0.305 to 1.11 ± 0.246 g/L) (*p* = 0.009) when compared to the values observed before consumption, similarly, to the intervention with orange juice (750 mL/8 weeks) [35] the population was divided in two group according to the body mass index: normal and overweight. Interestingly, in the overweight group TC concentration decreased before the consumption of orange juice (203 ± 40 to 188 ± 38 mg/dL) and LDL-C (138 ± 37 to 126 ± 37 mg/dL). However, in the normal weight group also TC decreased before the consumption (173 ± 20 to 159 ± 27 mg/dL) and LDL-C (104 ± 21 to 93 ± 22 mg/dL) (*p* < 0.05); without any alteration in body composition (body mass index, body fat mass, and waist circumferences) in either of the groups.

### 3.6. Study Quality Assessment

According to the risk of bias tool, there was an uncertain risk of bias in blinding (outcome assessment), allocation concealment, and selective reporting. We determined low risk of bias in incomplete outcome data and other source of bias. The item with more high risk of bias was blinding (Figure 2 and Appendix A). According with the quality of the reports, none of the studies reported all the elements of the CONSORT statement. The items that were least reported included method used to generate the random allocation sequence, type of randomization, and details of any restriction. The items that were reported for all the studies were specific objectives or hypotheses, eligibility criteria for participants, sources of funding and other support, role of funders among others (Appendix A).

## 4. Discussion

Nutrition plays a key role in prevention, control, prevention, and prognosis of non-communicable chronic diseases such as dyslipidemias. One of the most important characteristics of the treatment of dyslipidemias is the prevention of cardiovascular events, which depend on many factors such as the access to lipid-lowering therapies, including changes in diet. In this sense, the use of specific components such as isolated antioxidants compounds or modified foods emerges as a potential strategy for the reduction of the risk of dyslipidemias and CVD [37]. These bioactive compounds can be found naturally in a wide variety of foods; thus, it would be expected that the intake of these natural foods would be more complete and healthy than the consumption of single nutrients. Population of in middle-income countries presented health disparities and unhealthy eating patterns; thus, it is very common for inhabitants to use foods or herbs as accessible strategies to control different types of diseases. The present review summarizes the main features and findings of studies focusing on nutritional strategies exclusively including natural foods with antioxidant effects. This is important since natural food consumption not only provides bioactive compounds with antioxidant potential, but also nutrients such as fiber, carbohydrates, proteins, and minerals.

Interventions with almonds are associated with significant changes in the reduction of CT and LDL-C due to various bioactive compounds. According to the previous studies, the hypolipidemic effects of almond supplementation could be associated to their chemical composition. Almonds are rich sources of fiber, polyunsaturated fatty acids, proteins and amino acids, as well as vitamins, minerals, and polyphenols [20]. Polyphenols could modulate the expression of the genes of fatty acid synthase (FAS) and 3-hydroxy-3-methyl glutaric acyl coenzyme A reductase (HMGCR), which play important roles in lipid synthesis [38]. The molecular mechanisms by which polyphenols have an effect on lipids is related with the increase in energy expenditure, reduction in lipogenesis, reduction in fat mass and OS through activation of energetic sensors such as adenosine monophosphate (AMP)-activated protein kinase (AMPK) and sirtuin 1 (SIRT1). The activation of AMPK in a SIRT1-dependent fashion, in turns inhibits enzymatic activity of FAS, which is key downstream regulators of AMPK in the control of lipid metabolism, it reduces lipid accumulation, and increased fatty acid oxidation and/or decreased fatty acid synthesis [39].

Interestingly, important antioxidant and anti-inflammatory effects were observed in interventions containing nuts. In particular, nut consumption in humans can contribute to cardiovascular protection by improving lipid profiles, blood pressure, endothelial function and antioxidants. These effects have been attributed to high levels of unsaturated fatty acids and low levels of saturated fatty acids, as well as the presence of various polyphenols, carotenoids, and phytosterols [40]. The intervention with sesame seeds decreased TC and LDL-C levels and TC/HDL ratio; effects that were attributed to sesamin and/or episesamin lignans [28]. It has been reported that sesamin and episesamin promoted various physiological effects such as antioxidant [41], and lowering of blood pressure and serum lipids [42,43]. Some studies in animal models suggest a possible mechanism by which sesame seeds induce a hypocholesterolemic effect. The effect on TC is mediated by inhibition of the intestinal absorption of cholesterol and increasing the excretion of cholesterol as bile acids [42]. In addition, these compounds are also associated with the increased hepatic fatty acid oxidation through the activation of peroxisome proliferator-activated receptor alpha (PPAR-α), and decreased lipogenesis by down-regulation of sterol regulatory element binding protein-1 (SREBP-1), which in turns decreased serum lipid concentration [42,43]. In fact, the most valuable micronutrients found in cashews are folate and tocopherols, which delay metabolic disorders, protecting against atherosclerosis and other chronic non-communicable diseases (CNCD) [44].

Martínez-López et al. [27] reported a decrease in serum levels of TC, LDL-C, and TG after moderate coffee consumption (green (35%)/roasted (65%)), previous studies have shown that intake of green coffee bean extract for 28 days, providing 50 or 100 mg/day of chlorogenic acid, showed reductions in CT and LDL-C levels in hypercholesterolemic subjects [45]. Chlorogenic acid is active dietary polyphenol, with an important biological properties, the first is the antioxidant effects but there are positive effects on glucose and lipid metabolism regulation by activating the AMPK [46], up-regulating the expression of hepatic PPAR-α [47], and inhibiting the HMGCR [48]. In such a way that it can be postulated that chlorogenic acid can exert fundamental functions in the metabolism of glucose and lipids. In contrast to these results, there are other studies that associate coffee consumption with an increase in lipids, which is due to the presence of diterpenes (kahweol and cafestol) [49]. However, the concentration of these compounds is lower in instant and filtered coffee. Coffee is one of the most popular drinks that received considerable attention due to its potential antioxidant activity [36].

Furthermore, two of the included studies showed that other beverages also showed an antioxidant effect [34,35]. One of them showed that consumption of 750 mL of orange juice daily for 8 weeks decreased TC and LDL-C in serum. Orange contains bioactive compounds such as flavones, which have been attributed various beneficial effects [50]. Previous studies have shown that citrus juice intake as well as flavone supplementation can have an impact on TC by inhibition synthesis of TC and increased fatty acid oxidation [51,52]. Especially, vitamin C plays an important role in regulation of TC and TG concentration. In fact, low concentration of vitamin C is associated with the decreased cholesterol absorption due to low levels of bile acids, monoglycerides, and fatty acids [53]. On the other hand, Aghababaee et al. [34] found an increase in HDL-C after the intervention with blackberry juice, which contained anthocyanins. It has been demonstrated that anthocyanins generated many beneficial effects like hypolipidemic and antioxidants activities, inhibiting the inflammatory process and endothelial dysfunction [54]. Anthocyanins can act on inhibition of cholesterol ester transfer protein, whose function is transferring cholesteryl esters from HDL to Apolipoprotein B (ApoB) in exchange for triglycerides [55]. Furthermore, anthocyanins are also present in grapes, and therefore they could be mediating the similar effect observed with the intervention with red grapes, reducing TC [26].

In addition, the blackberry juice and raw almonds interventions showed a significant reduction in concentration of Apo-B100. Currently, it is known that lower concentrations of apolipoproteins ApoB and ApoA-I have been associated as stronger predictors of CVD than LDL-C and HDL-C. This can be explained due to the cholesterol enters the artery wall only if it is inside the ApoB particles. Therefore, the cholesterol will be deposited inside the artery wall is directly related to the amount of ApoB particles that are trapped inside the artery wall. Once inside the artery wall, the smaller ApoB particles have a greater tendency to become accumulating within the intimal space of the artery wall, resulting in the formation of atherogenic plaque [56].

In this sense, dietary strategies with food with antioxidant effects are gaining interest due to their promising therapeutic application in pathologies such as obesity, type 2 diabetes and its related complications [36]. However, most of studies include isolated antioxidant compounds or supplements, unlikely to be found in common diets. This is relevant, since natural foods not only provide antioxidants compounds, but they are also of great nutritional value due to their content of dietary fiber, protein, and minerals. The antioxidant activity provided by some food has an important effect in serum antioxidant activity and furthermore, it can act as a scavenger of ROS and inhibit oxidative damage. Antioxidant activities by consumption of different foods have also been associated with the presence of polyphenols, flavonoids, and carotenoids, which received considerable attention due to their potential antioxidant effect by the presence of hydroxyl groups and double bonds in chemical structure of polyphenols [36]. However, the antioxidant activity is not always increased by the consumption of antioxidant compounds present in foods or combinations of various antioxidant supplements. To promoted effects on circulating antioxidant activity is important to take into account the chemical disposition of antioxidant compounds presents in the food matrix as free, glycosylated or bound to other components. Nowadays, there are discrepancies about the effect of consumption of antioxidants from diet on antioxidant activity of plasma, and its physiological response [57]. Thus, it is important to mention that the antioxidant activity of plasma is results from the combination of dietary habits, environmental and physiological factors of each person. In fact, the antioxidant effect of bioactive compounds depends on bioavailability of dietary antioxidant compounds related with the mechanisms of digestion and absorption modulate by its own metabolism, thus these play a key role in the biological effects of the dietary compounds.

Interventions with sesame seeds, grapes, *G. paraguayense*, and granulated Brazil nuts decreased lipoperoxidation markers such as MDA and oxidized-LDL, suggesting a potential to directly neutralize ROS, and in turn to decrease lipoperoxidation. These foods contain a large range of phytochemicals with antioxidant properties like polyphenols, flavonoids, carotenoids, and ascorbic acid, which have been associated with potential health effects, and have proven their potential to decrease the metabolic alterations of certain chronic diseases such as cardiovascular disease, metabolic syndrome, and cancer [58,59,60,61].

Oxidative damage is directly related with increased ROS generation, and also to an imbalance in defense mechanism such as a decrease in antioxidants enzymes. Despite some evidence of the beneficial effect of different bioactive compounds in endogenous antioxidant system, the activity of antioxidant enzymes has barely been studied in humans. Here, we presented interventions in humans with *G. paraguayense*, aqueous wolfberry fruit, Brazil nut, and sesame seeds that promoted an upregulation on antioxidant enzymes such as CAT, SOD, and GPx. Although these studies did not propose a molecular mechanism to explain this effect, other studies have suggested that compounds present in foods like gallic acid, coumaric acid, kaempferol, and quercetin promote the activation of nuclear factor erythroid 2-related factor 2 (Nrf2), which resulting in the induction of endogenous antioxidant system. Nrf2 is a major regulator of phase II antioxidant responses, induced the expression of numerous genes of protective enzymes and scavengers. In normal conditions, Nrf2 is bind to Kelch-like ECH-associated protein 1 (Keap1) in cytoplasm of cells; however several stimuli, including oxidative stress and antioxidants compounds, lead to the disruption of Nrf2/Keap1 complex, and then Nrf2 can be translocate to the nucleus where it binds to specific sequence of DNA call antioxidant response element (ARE) [62].

The increase of antioxidant enzymes by dietary interventions in population with dyslipidemias is of great relevance, since these are the first line of defense against the increase of ROS. Epidemiological studies have shown that during metabolic alterations there is an inverse relationship between CVD with circulating levels of SOD, CAT, and GPx activities [63]. However, the antioxidant activity of these enzymes also depends on the bioavailability of other elements such as selenium. Selenium is an important component incorporated to selenoproteins involved in enzymatic functions of antioxidant and anti-inflammatory, thus it may play an important role in the protection against OS via selenoproteins as GPx [64]. In fact, low concentration of selenium results in decreased GPx protein levels and in its enzymatic activity [65]. In this way, we can mention that selenium-dependent systems have a critical rol in the antioxidant defense in humans.

Another important endogenous antioxidant is GSH, which acts a substrate for the antioxidant enzyme GPx. GSH is a non-enzymatic antioxidant that plays an important role in detoxification through the formation of reactive metabolites, and also protects cell membranes against oxidation. Concentration of GSH is decreased in individuals with vascular disease, which is a complication generated by dyslipidemias [5]. Thus, after intervention with *G. paraguayense* or Mate tea, GSH concentration significantly increased that could be associated with phenolic antioxidant compounds present in these types of foods [24,29]. Some studies have been shown that increased GSH concentrations was directly associated with a higher consumption of foods rich in polyphenols, vitamin C, and fiber [66,67].

Furthermore, others non-enzymatic antioxidants are provided by lipid-soluble and water-soluble antioxidants. Thus, these bioactive compounds such as ascorbic acid and α-tocopherol could act by several mechanisms including scavenging of ROS [68]. An intervention with *G. paraguayense* [24] showed increased concentrations of vitamin C and E, thus the synergistic interaction between vitamins C and E may be critical at early stages of atherosclerotic lesion formation by neutralizing and scavenging ROS. Additionally, the antioxidant effect of these vitamins are involved in the preservation of integrity of the cell membrane, thereby potentially interfering with the pathogenesis of atherosclerosis [69,70].

Moreover, the revised studies provide evidence about the anti-inflammatory effect of foods with antioxidants, which can be directly attributed to scavenging of ROS and/or to the suppression of pro-inflammatory signal transduction. Antioxidant effects of the dietary interventions are of great importance in dyslipidemias, since OS is the main cause of vascular endothelium damage, which leads to endothelial dysfunction underlying the development of atherosclerosis [5]. The studies selected showed a significant reduction in inflammation markers such as TXB2, VCAM-1, and CRP by consumption of olive oil, walnuts intake, orange, and blackberry juices, respectively. Antioxidants such as resveratrol, oleuropein, hydroxytyrosol, caffeic acid, and anthocyanidins provide anti-inflammatory effect throughout neutralization of ROS. Excessive levels of ROS can induce activation of mitogen-activated protein kinases (MAPK) that in turns, cross talks with other pathways such as nuclear factor kappa-light-chain-enhancer of activated B (NF-κB). NF-κB is a complex protein that plays a key role in transcription of the expression of a large number of genes involved in inflammation such as pro-inflammatory cytokines including IL-1, IL-2, IL-6, and TNF-α, chemokines, adhesion molecules, immune-receptors, growth factors, and other agents involved in proliferation and invasion [71]. Its activation by ROS is through phosphorylation of I-κB proteins by upstream kinases, which lead to the ubiquitination and degradation of I-κB. Active NF-kB then translocates into the nucleus and activates the target genes, leading to increase in inflammatory transcription factors that up-regulate pro-inflammatory mediators [72]. Therefore, dietary interventions that promote the regulation of the inflammation process in dyslipidemias have a great importance since through this compensatory response the injuries caused by the inflammatory molecules. In addition, this effect also can result in decreasing the levels of ROS, and the harmful reactions caused by the chronic changes caused by oxidized lipids, and in turns decreasing the formation of atherosclerotic lesions. Thus, several studies have been shown the importance of the consumption of foods with antioxidants compounds due to its beneficial effects such as vasodilatatory, anti-platelet aggregation, and anti-inflammatory. These effects are associated with the chemical composition mainly characterized by its high content of a series of polyphenols compounds. These bioactive compounds generated anti-inflammatory and antioxidant effects by down-regulation of MAPK and NF-κB signaling pathways and induction of Nrf2, controlling the production of many inflammatory mediators and activation of endogenous antioxidant system [73].

It is important to consider that not only the antioxidant and anti-inflammatory effects of the consumption of this type of food with antioxidant effect should be highlighted. Since there are another risk factors that can be regulated in dyslipidemias including body weight and amount of adipose tissue and its distribution. These are factors that are related to the development of hypertension, insulin resistance and CVD during dyslipidemia [74]. Studies have shown that weight reduction improves lipid and glucose metabolism, and it is associated with the reduction in adipose tissue [75]. In addition, it is very important to take into account the sedentary lifestyle also is associated with increase in the risk of CVD by exacerbating insulin resistance and dyslipidemia. A meta-analysis has shown the favorable metabolic effects of exercise in adults with hyperlipidemia by reduction in body weight and adipose tissue [76].

A potential limitation of the present revision is the fact that the search was limited to a few databases (PubMed and Scopus, and Web of Science); and only the English language, however a large number of studies were identified. Nevertheless, given that the included studies provide supporting evidence of potential benefits for dyslipidemias by the intake of natural food with antioxidant effect, these results may be taken into consideration when planning public nutritional strategies relying on foods that are accessible to the population and are in accordance with their lifestyle, social life, and eating habits. Moreover, these strategies would have the advantage of not including modified products that are not only difficult to obtain but also imply a higher economic cost. This information might well be considered for developing policies regarding public health, particularly in those countries where dyslipidemias are of high prevalence, such as Mexico and other Latin American countries [77].

## 5. Conclusions

In conclusion, this systematic review compiling scientific about dietary strategies that used foods with antioxidant effect to decrease oxidative damage and inflammation to improve the antioxidant status in dyslipidemias. The sixteen reviewed studies suggest that the mechanism exerted by antioxidants food to protect subjects with dyslipidemia against the associated complications include modulation of lipid homeostasis, antioxidant activity, and anti-inflammatory processes. Results might be of potential interest for the planning of nutritional strategies in dyslipidemias.

## Figures and Tables

**Figure 1 antioxidants-10-00225-f001:**
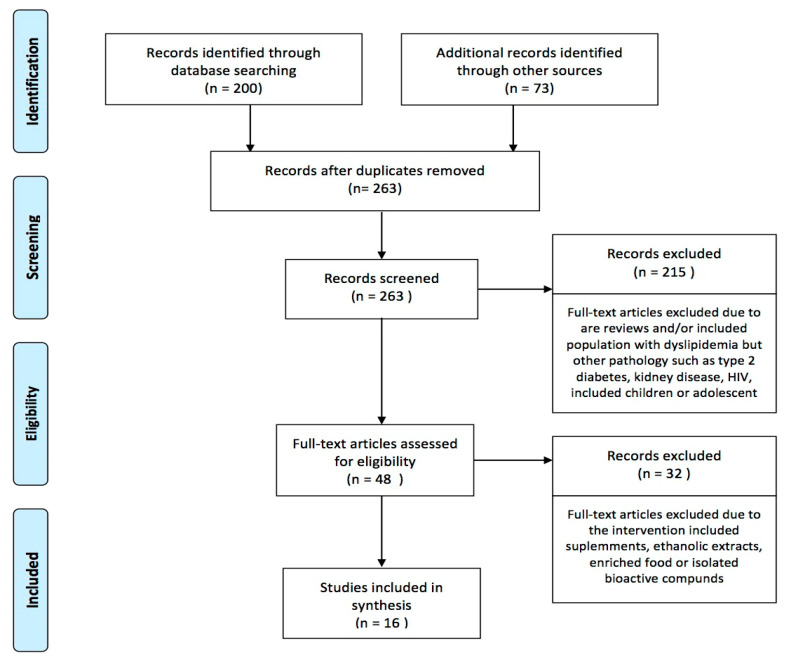
Flow chart (PRISMA) of studies included.

**Figure 2 antioxidants-10-00225-f002:**
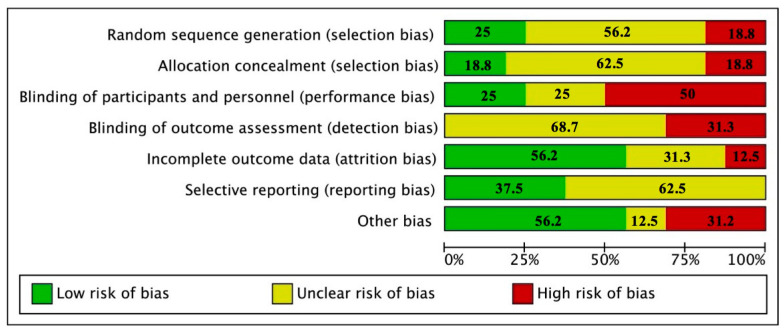
Report of the study quality index components in studies included.

**Table 1 antioxidants-10-00225-t001:** PICO criteria for criteria for study selection.

Criterion		Description
**P**	Population	Subjects with dyslipidemias
**I**	Intervention	Dietary interventions including foods with antioxidant effect
**C**	Control	Any comparator or comparison Placebo Control diet Without comparator
**O**	Outcomes	Blood lipids Antioxidant activity Oxidative and inflammatory markers

**Table 2 antioxidants-10-00225-t002:** Characteristics of the studies including in systematic review.

First Author, Year	Location	Study Population (Metabolic Condition, Sample Size)	Study Design	Dietary Intervention	Follow-Up	Significant Findings
Ros, 2004 [21]	Spain	Moderate hypercholesterolemia (n = 21)	Randomized, crossover trial	Intervention with Mediterranean-type diet (control group) and with isoenergetic diet with walnuts (intervention group)	4 weeks without washout period	Walnut reduced TC, LDL-C and VCAM-1
Visioli, 2005 [23]	Italy	Mildly dyslipidemia (n = 22)	Randomized, crossover trial	Intervention with (40 g/d) of Extra-virgin olive oil (EVOO) or Refined olive oil rich (ROO)	7 weeks with 4-week washout period between intervention	EVOO decreased serum thromboxane B2 and increased plasma antioxidant capacity
Jalali-Khanabadi, 2010 [20]	Iran	Mild hyperlipidemia (n = 30)	Baseline and post intervention effects	Intervention with raw almonds (60 g/d) 2 times daily	4 weeks	Raw almond significantly decreased TC, LDL-C and Apo-B100
Yu-Ling, 2011 [24]	China	Hypercholesterolemia (n = 18)	Baseline and post intervention effects	Intervention with *G. paraguayense* (100 g/d)	8 weeks	*G. paraguayense* increased ascorbic acid, α-tocopherol, glutathione, GPx and CAT activities in erythrocyte; and decreased MDA
Alipoor, 2012 [28]	Iran	Hyperlipidemia (n = 38)	Randomized trial	Interventions with white sesame seeds (40 g/d) or control diet without white sesame seeds	60 days of intervention	White sesame decreased serum TC and LDL-C and lipid peroxidation; and increased activities of GPx and SOD
Boaventura, 2012 [29]	Brazil	Dyslipidemia (n = 74)	Randomized trial	Interventions with Mate tea (MT) (330 mL, three times at day) or Dietary intervention (DI) or MT + DI (330 mL, three times at day)	90 days of intervention	All interventions significant increase antioxidant activity and glutathione concentrations
Berryman, 2013 [22]	United States of Amercia	Moderate hypercholesterolemia and overweight (n = 15)	Randomized, 4-period, crossover trial	Interventions with whole walnuts (85 g) or skinless walnuts (34 g) or oil from skinless walnuts (51 g) or walnut skins (5.6 g) once time	Once time and with 1-week washout period between intervention	Walnut skins decreased the reactive hyperemia index and increased antioxidant potential
Carvalho, 2015 [32]	Brazil	Dyslipidemia and hypertension (n = 77)	Randomized, placebo-controlled, double-blind trial	Intervention with Brazil nut flour (13 g/d) or placebo which consist in dyed cassava flour (11 g/d)	90 days	Brazil nut flour reduced TC, non-HDL-C and Apo A-I.
Apostolidou, 2015 [25]	Greece	Hypercholesterolemia (n = 20) and normal cholesterol (n = 17)	Randomized, crossover, placebo-controlled trial	Intervention with red wine or placebo (red colored water with 1% of ethanol) (125 mL/d and 250 mL/d)	1 month and 1-month washout period	Red wine reduced the risk factors for cardiovascular disease by increasing TAC and a-tocopherol (vitamin E)
Huguenin, 2015 [33]	Brazil	Dyslipidemia and hypertension (n = 91)	Randomized crossover trial, double-blind, placebo controlled trial	Intervention with granulated Brazil nut (13 g/d) or placebo (flavored cassava flour) (10 g/d)	12 weeks and 4-week washout period between intervention	Granulated Brazil nut intervention significantly increased plasma selenium, GPx activity, and reduced oxidized-LDL
Rahbar, 2015 [26]	Iran	Hypercholesterolemia (n = 69)	Randomized trial	Interventions with Condori red grapes (500 g/d) or Shahroodi white grapes (500 g/d) or control (500 g/d of fruits, except grapes)	8 weeks	Red and White grapes reduced lipoperoxidation, TC and LDL-C and increased total antioxidant capacity
Dourado, 2015 [35]	Brazil	Normal weight (n = 25) and Overweight (n = 25)	Baseline and post intervention effects	Orange juice (750 mL/d)	8 weeks	Orange juice reduced in TC, LDL-C, CRP and lipid peroxidation; and increase in IL-12 and TAC in both groups
Aghababaee, 2015 [34]	Iran	Dyslipidemia and hypertension (n = 72)	Randomized trial	Intervention with blackberry juice with pulp (300 mL/d) or control group (usual diet)	8 weeks	Blackberry juice increased Apo A-I and HDL-C and decreased Apo B100 and hs-CRP
Lee, 2017 [30]	Korea	Mildly hypercholesterolemia and overweight and (n = 53)	Randomized double-blind trial	Intervention with aqueous extract of Wolfberry fruit (13.5 g in 80 mL/d) or placebo (80 mL/d)	8 weeks	Aqueous extract of Wolfberry fruit decreases in erythrocyte superoxide dismutase and catalase activity and DNA damage in lymphocytes
Chiu, 2017 [31]	China	Mildly hypercholesterolemia (n = 44)	Randomized, double-blind, placebo-controlled trial	Intervention with 60 mL/day of fermented beverage made from fruits and vegetable or placebo extract	8 weeks	Fermented plant extract decreased the anthropometric parameters, TC, and LDL-C, and remarkably elevated TAC and LDL oxidation
Martínez-López, 2019 [27]	Spain	Hypercholesterolemia (n = 27) and normal cholesterol (n = 25)	Randomized, crossover, controlled trial	Intervention with coffee or control beverage (water or an isotonic drink, caffeine and polyphenol free) (200 mL three times a day).	8 weeks and 3-week washout period between intervention	Coffee promoted a reduction in TC, LDL-C, and TG, only in subjects with hypercholesterolemia. Also, in both groups improved TAC and decrease of body weight, blood pressure, heart rate and MDA concentration

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
