# Peer review of "Dietary Strategies by Foods with Antioxidant Effect on Nutritional Management of Dyslipidemias: A Systematic Review"

_antioxidants, 2021, doi:10.3390/antiox10020225_

Round 1

Reviewer 1 Report

Alhough this results are interesting for the planning of nutritional strategies in dyslipidemias, but I believe that it can be accepted after minor improvement. In whole manuscript a lot of shorthand is used, what is misleading.

Some issues need to be clarified or added on the manuscript before it can be published:

  1. please provide research hypotheses, since you refer to them later
  2. from line 147 to line153 remove to methods
  3. line 181 ROS explain reactive oxygen species
  4. line 182 CVD explain cardiovascular disease
  5. could you at the end of the article create a table explaining all the abbreviations
  6. it is worth adding information on cashew nuts, for example: The most valuable micronutrients found in cashews are folate and tocopherols, which delay metabolic disorders, protecting against atherosclerosis and other chronic non-communicable diseases (CNCD).

 Dias CCQ, Madruga MS, Pintado MME, Almeida GHO, Alves APV, Dantas FA, et al. (2019) Cashew nuts (Anacardium occidentale L.) decrease visceral fat, yet augment glucose in dyslipidemic rats. PLoS ONE 14(12): e0225736. https://doi.org/10.1371/journal.pone.0225736

  1. line 189 skin nuts what did you mean
  2. line 190 what kind of fruits and vegetables (beverage made from) explain
  3. line 236 what kind of plants explain

Reviewer 2 Report

In this in-depth systematic review the authors have tried to assess which dietary strategies using foods with antioxidant effect to improve  the antioxidant status in dyslipidemia. The work of assessing the literature on the subject has been done in a timely and comprehensive manner.

Abnormal diet is considered to be an important risk factor for dyslipidemia and in recent years many associations between lifestyle and improvement of the patients' lipid profile have been demonstrated.

In this paper the authors have placed a lot of attention on the possible correlation between the antioxidant power of the individual food and the improvement of dyslipidemia. Although metodologically correct, this reductionist approach to nutrition has shown great limitations. For example, demonstrating that the consumption of nuts has a positive effect on the health of the individual risks may not take into account everything else: weight, abdomen circumference, physical activity, etc.

The authors should therefore review the discussion of the paper. The list of all the conclusions of the articles under consideration makes it unclear how useful this review is for clinical practice and for the development of further literature on the subject. The authors should emphasize the salient aspects of the articles trying, also through the use of a summary table, to show a more global approach on the possible benefit of the consumption of some foods for the improvement of the lipid balance of the subjects.

Reviewer 3 Report

I appreciate this invitation to review manuscript by Isabel Medina-Vera et al. entitled: “Dietary strategies by foods with antioxidant effect on nutritional management of dyslipidemias: A Systematic Review" submitted to Antioxidants.

Isabel Medina-Vera et al. performed a very elegant review regarding the antioxidant effects of nutritional strategies by natural foods with antioxidant properties on blood lipids, enzymatic and non-enzymatic antioxidants and oxidative and inflammatory markers of subjects with dyslipidemia, but without type 2 diabetes, kidney disease, and/or human immunodeficiency virus. After careful searching of MEDLINE/PubMed, Scopus and Web of Science databases, and screening of 263 studies, 16 of them were included. Authors summarized that dietary strategies which include walnuts, olive oil, raw almonds, G. paraguayase, white sesame, mate tea, Brazil nut flour, red wine, granulated Brazil nuts, grapes, wolfberry fruit, fermented beverage, coffee, orange and blackberry juices show the significant effects on blood lipids, antioxidant activity, antioxidant enzymes and oxidative and inflammatory markers. The article is quite well written, a study design is quite well planned, and results show interesting findings. I do not have any concerns about this paper.

Reviewer 4 Report

Very good and followed the basic guidelines for a systematic review 

Round 2

Reviewer 2 Report

The authors greatly improved the paper following my suggestions.